# Challenges in the Computational Modeling of the Protein Structure—Activity Relationship

Gabriel Del Río 

Department of Biochemistry and Structural Biology, Institute of Cellular Physiology, UNAM, Mexico City 04510, Mexico; gdelrio@ifc.unam.mx

**Abstract:** Living organisms are composed of biopolymers (proteins, nucleic acids, carbohydrates and lipid polymers) that are used to keep or transmit information relevant to the state of these organisms at any given time. In these processes, proteins play a central role by displaying different activities required to keep or transmit this information. In this review, I present the current knowledge about the protein sequence–structure–activity relationship and the basis for modeling this relationship. Three representative predictors relevant to the modeling of this relationship are summarized to highlight areas that require further improvement and development. I will describe how a basic understanding of this relationship is fundamental in the development of new methods to design proteins, which represents an area of multiple applications in the areas of health and biotechnology.

**Keywords:** protein structure; protein function; bijection

## 1. Proteins: Fundamental Polymers for Biological Systems

Biological systems are composed of biopolymers, including nucleic acids, polymers (e.g., DNA, RNA), carbohydrates (e.g., starch, cellulose), lipids and proteins. All these biopolymers play critical roles in all the processes associated with life. Biopolymers are composed of monomers that are linked in a sequential way; for instance, proteins are composed of amino acids, while nucleic acids are compose of nucleotides and polymers of carbohydrates are composed of different carbohydrate monomers (e.g., starch and cellulose are made of glucose mainly), and a similar situation occurs with lipids. Since each biopolymer is composed of many different monomers, this situation gives rise to an enormous number of different biopolymer sequences. For instance, proteins are composed of 20 different amino acids; hence, proteins that include up to 100 monomers of amino acids will encompass $20^{100}$ different protein sequences.

The information about the state of the system is maintained by these biopolymers, and this is achieved by controlling the presence/activity of these biopolymers. Activity refers to the action that these biopolymers perform, either maintaining a particular morphology, transporting cargo or altering the chemical structure of other molecules, among others. This activity is commonly referred to as function. In this review, I will use the term activity to avoid confusion with the mathematical term function, which I will also use in this review. The flux of information in biological systems that involves all these biopolymers is depicted in Figure 1, where proteins are proposed to play a central role. The central dogma in molecular biology [1] is depicted at the bottom of the figure, where the information flux goes from DNA to RNA (transcription) and from RNA to proteins (translation). The figure includes bidirectional arrows to indicate that the information also goes from RNA to DNA (retro-transcription) and from protein to RNA or DNA (epigenetics); the information flux may go from DNA to DNA, from RNA to RNA or from protein to protein, as indicated by circular arrows. The other biopolymers (carbohydrates and lipids) are inserted into this flux of information in the top part of the figure. It is relevant to note that proteins participate in any of these fluxes, e.g., protein transcription factors are required for tran-

scription, ribosomal proteins are involved in translation, and the synthesis/degradation of carbohydrates or lipids also requires proteins.

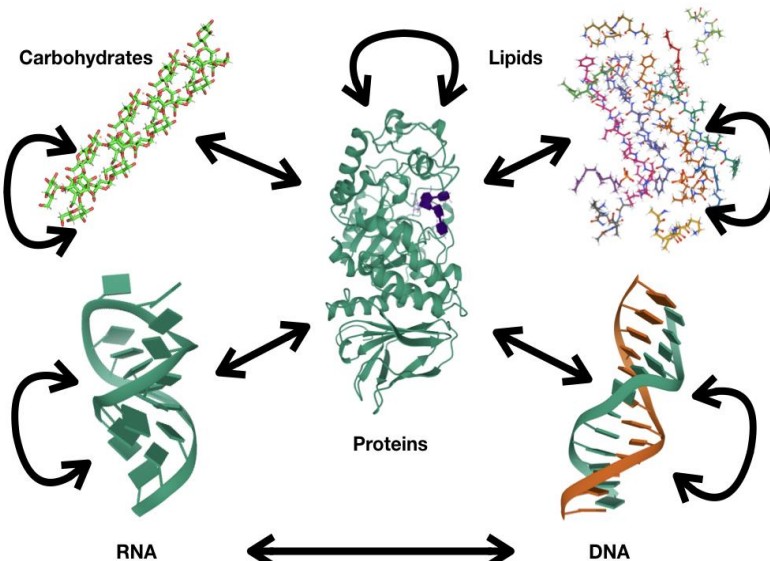

**Figure 1.** Information flux among biopolymers.

Thus, proteins are central to life due to the different activities that they perform in biological systems. In consequence, it is expected that knowledge of the protein contents of living organisms may help to anticipate the capabilities of such systems. For instance, having access to the list of protein sequences in a tumor would provide the information required to anticipate the tumor's malignancy; knowing the proteins of viruses may anticipate their virulence. Due to the advances in DNA sequencing techniques, having access to protein sequence data is relatively straightforward nowadays. However, knowledge of protein sequences is not enough to anticipate the activities that proteins may display in biological systems. This is because, at the most basic level, protein activity depends on the ability of proteins to interact with other molecules; hence, protein activity depends on the context in which the protein is located. This dependency of proteins was originally referred to as gene sharing, but, more recently, it has been considered a particular case of a phenomena referred to as moonlighting [2]. For instance, when a protein referred to as thymidine phosphorylase is present inside human cells, at the cytoplasm of these cells, it catalyzes the dephosphorylation of thymidine, among other nucleotides; yet, when this same protein is expressed outside of cells, it stimulates cell growth and chemotaxis [3].

From the previous notion, it can then be surmised that different contexts provide proteins with different molecules to interact with, and this may be in turn translated into different protein activities. The ways in which this differential binding translates into different activities highlights another feature of proteins: allostery [4]. Originally proposed by Monod and Jacob in 1961, the model nowadays refers to conformational changes in any protein induced by binding to other molecules, which are required for the performance of a particular activity [5]. Take, for instance, the case of hemoglobin (Hb), a protein carrying two oxygen molecules that are distributed from blood to tissues, which displays several forms of allostery: (i) intrinsic, when a single oxygen molecule binds to Hb and the binding of the second oxygen molecule is facilitated or (ii) heterotrophic ligands (2,3-biphosphoglycerate, protons, carbon dioxide, among others) facilitate the binding of both oxygen molecules [6]. Another important aspect that controls protein activity are post-translational modifications; these are modifications to the chemical structure of proteins (e.g., addition of a phosphate group, addition of a carbohydrate) that change the protein activity, tag the protein to be degraded or change the localization of a protein, otherwise changing the environment of the protein [7]. These features highlight an aspect of the

relationship between protein sequence (as encoded in DNA) and protein activity: this relationship is redundant because, for the same protein sequence, there are many different protein activities.

## 2. Protein Structure

It is generally accepted that proteins adopt a conformation in the three-dimensional space. Such conformation is commonly referred to as the protein structure. In this structure, some elements are observed that are frequently found in proteins: alpha-helixes, beta-sheets, kinks, among others. These elements are commonly referred to as secondary structures. All these elements are derived from distinct patterns of turns in the protein backbone [8]. As noted above, a single protein sequence may adopt multiple conformations or structures, but it was originally recognized that these conformations would not be too different. For instance, a protein will adopt a particular protein structure that display features that are recognizable, such as helixes, in a particular order (bundles of six helixes), that may change the orientation between them or one of the features may disappear, but the overall features are conserved (see Figure 2).

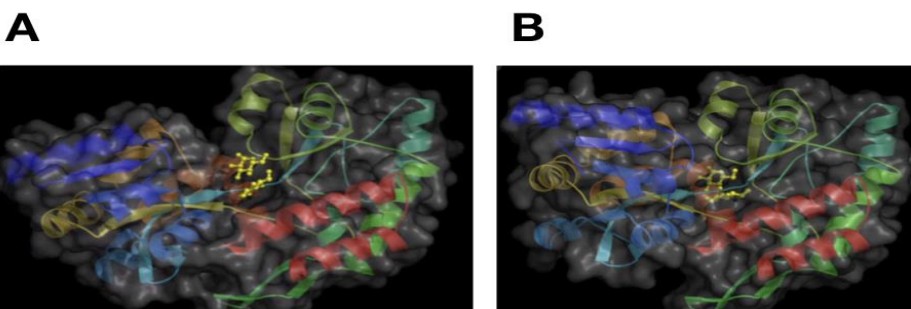

**Figure 2.** Maltose-binding protein motion. A representation of the maltose-binding protein bound to maltose (yellow object in the center of the image) is presented. The surface accessible to water is presented in gray, and secondary structural elements (helixes and sheets) are presented as ribbons of different colors (blue, cyan, greens). Image in (**A**) shows the "open" conformation of the protein and (**B**) the "closed" conformation of the protein. Please note that the same secondary structural elements are present in both images, but the orientation between these changed. Pictures were obtained from the database of macromolecular motions [9].

This notion was derived from the original observations that protein sequences will fold to reach a final conformation that has one minimum of energy. However, this notion has been changed by recognizing that a single protein sequence may adopt multiple structures as a consequence of finding multiple energy minimums [10]. Examples of this new paradigm are natively unfolded proteins and metamorphic proteins. Metamorphic proteins drastically change their structure when the environment changes [11]; an example of this class of proteins is the Prolyl tRNA synthetase from *Plasmodium falciparum*, which, upon binding to the veterinary medicine halofuginone, switches from a helical structure to a beta structure [12]. Natively unfolded proteins or intrinsically disordered proteins (IDPs) have the ability to adopt multiple structures at binding to different partners or in different environments, and, even in the bound state, these proteins remain disordered [13]. IDPs are commonly found in the transmission of information between proteins, a process known as signal transduction [14].

All this evidence indicates that proteins exert their activity by adopting different structures. It may be argued that an ensemble of protein structures is associated with one activity because this set of proteins structures are required to bind the multiple conformations available to the binding molecule. In agreement with this notion, it has been noted that the structure–activity relationship presents both injective and surjective features; hence, this relationship should be a bijection [15]. That is, the structure–activity relationship in a protein is a one-to-one relationship, which differs from the redundant nature of protein

sequence and activity relationship. To learn how this relationship operates, it is important to have access to many instances of protein structures bound to different molecules. At the time of writing this review, less than 180,000 protein structures were available in the protein data bank [16], derived from less than 30,000 different protein sequences as inferred from the UniProt database [17], and less than 900,000 experimental activity annotations were reported for all known genes [18]. However, this limitation may change considering the recent solution reported in predicting protein structures from protein sequences in the 14th edition of the Critical Assessment of Protein Structure Prediction, CASP [19]. Nowadays, for every protein structure, there are more than 10 protein sequences known for which the structure is unknown. Thus, we expect that protein structure models will further enhance our understanding of protein structure–activity relationships.

### 3. Modeling Protein Structure–Activity Relationship

From a mathematical perspective, the structure–activity relationship may be represented by a mathematical function if structure (S) and activity (A) are represented by numerical values:

$$A = f(S) \tag{1}$$

S is commonly represented by a matrix of size $3 \times N$, where N is the number of atoms in the protein and the 3 corresponds to the Euclidean coordinates in x, y and z of every atom in that protein. However, S may be represented by vectors as well [20–22]. Activity, on the other hand, is commonly represented by a single value (e.g., binding affinity expressed in energy units) or by a hierarchical numeration (e.g., gene ontology classification [23]). Alternatively, protein activity may as well be represented by a matrix (e.g., protein contact map [24]). There is no standard way to represent these two features and, consequently, there is no standard way to establish the functions relating to these protein features. The Critical Assessment of Functional Annotation (CAFA) challenge may provide the grounds to standardize these representations and promote the development of better models to predict protein function from protein structure [25]; however, CAFA is based on protein sequences due to the limitation of available protein structures, a scenario that, we expect, may change in the coming years.

Since the mathematical function relating protein structure and activity is not known, it is possible to use data to learn it. A computational approach to achieve this is machine learning, which may be used to approximate a solution:

$$g(S) = f(S); g(S) = A \tag{2}$$

where g(S) is the function learned with machine learning approaches that relates the protein structure (S) with protein activities (A). Both structure and activity are features derived from the same object and consequently can be assumed to be related. However, protein structure models and protein activity models are not necessarily related; for instance, a matrix representation of proteins structure is not necessarily related to a numeric value measuring the catalytic constant of an enzyme or the text describing an activity. Thus, it is convenient to have representations for both protein structure and activity that facilitate the study of this relationship at the model level.

One way to achieve this is by considering that protein activity may be inferred by predicting the binding partners of proteins [26]. For instance, a protein binding to a carbohydrate may provide an initial clue about its activity; proteins binding to other proteins that are located in the nucleus also provide further insight into protein localization. Different computational approaches have been described to predict protein-binding partners that may be divided into: (i) physics-based approaches [27], (ii) machine-learning-based approaches [28] and (iii) phylogeny-based approaches [29].

Another way to infer protein activity is through the prediction of critical sites; these sites include the amino acid residues that are critical for the activity of the protein. These sites may be predicted from the protein sequence or protein structure [30]; in this last case,

the critical residues may be related to central residues from the protein contact map connectivity [31], an aspect that is relevant for protein structure prediction, as I will describe later. This set of critical residues may be represented by a vector; hence, Equation (1) can be represented as follows:

$$C = f(S) \tag{3}$$

Since S is actually an ensemble of protein conformations, C may be represented also by a set of vectors. Thus:

$$\mathbf{C} = f(\mathbf{S}) \tag{4}$$

where $\mathbf{C} = \{C_i, ..C_j\}$ and $\mathbf{S} = \{S_i..S_k\}$. $\mathbf{S}$ only refers to the protein conformations associated with the protein activity represented by the $\mathbf{C}$ set. The nature of this relationship implies that S and C come in pairs: $\{S_i, C_i\} \ldots \{S_j, C_j\}$. That is, for every functional conformer of a protein, there is a set of critical residues associated with it. It would be expected that no identical set of critical residues will be found for different functional conformers of proteins; this is to maintain the proposed bijective relationship of protein structure with protein activity. Accordingly, it has been observed that different protein conformations observed crystallographically or computationally generated do not have fully overlapping sets of central residues [32] and presumably of critical residues for protein activity.

It is important to keep in mind, though, that protein binding does not establish the protein activity; for instance, a protein binding DNA may promote transcription or may act as an inhibitor of transcription. Finding the mathematical function to relate protein structure to protein activity (see equations above) remains an open question that requires further protein structural data and further experimental characterization of protein activities in different contexts. To describe the current challenges in modeling protein structure and activity, three of the best methods to model different aspects of this relationship are summarized.

## 4. Databases and Prediction Methods

There is one public repository that includes protein sequences, structures, modifications (post-translational or mutations), cellular localization and activity annotations: UniProt. This database is accessible through a website and is updated twice a year. In the last UniProt release, there were 1,049,123 sites (167,217 active sites, 405,245 metal binding sites and 415,853 binding sites) annotated for 214,812 UniProt entries. Despite the accumulation of information about proteins, this information is not readily available for predictors to develop models for the protein structure–activity relationship. I will summarize some of the most recent works in this area and compare the dataset used in these studies to exemplify this limitation. It is important to note, though, that a recent report included the largest comprehensive mutagenesis data on 14 proteins, which is a starting point to build a proper dataset to predict critical residues [33]. This review is not intended to summarize all different approaches reported so far to model protein activity that has been reviewed elsewhere [34]; this review focus on describing some aspects that are fundamental in using protein structure to model protein activity.

## 5. Predicting Critical Residues for Protein Activity

CsmetaPred [35] is a consensus method for predicting catalytic residues; these residues are a particular class of critical residues. Catalytic residues are relevant for proteins that accelerate a chemical reaction; such proteins are referred to as enzymes. Another important feature of catalytic residues is that these bind to substrates, the molecules that enzymes act on. Critical residues do not always bind to a molecule: they may be relevant to maintain protein structure or pH or some other features relevant for the protein activity. CsmetaPred uses four active site prediction methods, including CRpred, CATSID, DISCERN and XIA2. These methods use sequence and/or structural features (phylogeny, structure, physicochemical descriptors) and statistical and machine-learning approaches; thus, it represents a state-of-the-art predictor in this task. A z-score is obtained from

the four predictors for each residue, and then an average residue score is used to rank residues. To test this meta-predictor, the authors compiled six datasets that included a total of 1,619 proteins (protein structures) and an average of three catalytic residues per protein; therefore, most residues were annotated as non-catalytic. To deal with this imbalance in their dataset, the authors evaluated their predictions using the area under the positive rate curves and mean average precisions. The authors showed that CsmetaPred outperformed any of its four individual predictors and successfully predicted all known catalytic residues for 73% of the tested enzymes.

Critical residues are identified commonly by mutagenesis; that is, an amino acid is substituted at a given protein's position and its effect on the protein's activity is measured. It is common to assume a binary effect on protein activity for amino acid substitutions: there are substitutions that do not affect the protein activity and those which do affect it. This assumption is convenient because it can be used on any protein's activity, and it does not depend on a specific aspect of the activity that is measured. Provided that each mutation can be annotated as active or not active, a particular position may have multiple amino acid substitutions that affect protein activity and other substitutions that do not. One possible assumption is to use the number of non-tolerated substitutions as a criterion to rank critical residues [36]; another possibility is to consider the sequence conservation of the non-tolerated substitutions to rank critical residues [37]; using a combination of scores has also been shown to render good efficiency to identify critical residues [31]. Therefore, different annotation strategies of the ground true to be modeled render different efficiencies to predict critical residues. Having a common dataset to train methods aimed to predict critical residues will facilitate the evaluation of the different strategies to annotate critical residues.

This, in turn, will lead to unifying the dataset of mutants and annotations of critical and functional sites. To this end, it will be useful to consider the definition of protein activity used by the Gene Ontology Consortium [23], which includes three classes: molecular function (this is what I refer here to as protein activity), biological process (the context in which the protein performs its activity) and cellular localization. Currently, the interpretation of protein mutations is based on the molecular function or biological process of the protein and, consequently, predictors are biased to identify critical residues for these protein activities. The basis for this bias is that mutagenesis experiments of proteins require the analysis of thousands of protein variants, and the experimental determination of molecular function or a phenotype associated with the biological process is usually easier to detect than the localization of the protein; for instance, a protein that is relevant to the biological process "cell cycle" may display a phenotype where cells may not grow. Recent advances in fluoresce tagging to experimentally localize proteins [38] may facilitate estimation of the effect on cellular localization of protein mutations.

## 6. Prediction of Protein Activity

NetGO [39] is one of the best methods described to predict protein activity from protein sequences. The method is inspired by a multi-label classification problem where one protein sequence has multiple activity annotations (gene ontology terms). Since protein activity depends not only on the sequence but on the environment and the protein interactions, NetGO uses protein–protein interaction data annotated in the STRING database; the STRING database includes physical and functional protein interactions [40]. To accelerate the annotation of protein activity, NetGO uses a learn-to-rank method (LambdaMART); this method is a pairwise classification approach where, for every pair of GO terms associated with a given protein, the method ranks the GO terms rather than producing a score for each term and protein. In brief, NetGO uses protein sequence information and protein–protein interactions, combined with sequence comparisons and machine learning algorithms, to predict protein activities. The last results of the third CAFA contest report that NetGO achieved Fmax scores of 0.62, 0.4 and 0.61 for predicting molecular function, biological process and cellular localization, respectively [25].

A particular problem for protein activity predictors involves cases where a single-domain protein possesses multiple activities; this is the case of moonlighting proteins. Although there are predictors of moonlighting proteins [41,42], none of the proteins in the MoonProt database [43] were identified by a prediction, but through serendipity. The last release of the MoonProt database includes information about protein structure, hence facilitating the exploration of protein structure–activity relationships for this class of proteins.

Few servers are available that predict protein activity from protein structure that have been recently reviewed [44]. However, at the time of writing this review, these were not operating. This represents an area of opportunity, especially after this last year, when predicting protein structure from sequence achieved reliable results (see below).

## 7. Prediction of Protein Structure from Protein Sequence

Prediction of protein structure from protein sequences represents a means of modeling the protein sequence–structure–activity relationship. In the 14th edition of the CASP contest, AlphaFold2 achieved a score that matched the error observed in experimentally determined structures [45]. While the method used by AlphaFold2 is not public yet, it may be an extension of the original AlphaFold [46]. It has been shown that the use of full true contact maps or partial true contact maps of residues to predict structural models of proteins using heuristic algorithms achieved very accurate models [47], despite the fact that such a problem has been recognized to be NP-hard (non-deterministic polynomial-time hardness) [48]. These previous studies encourage the development of AlphaFold, which is a "distance map predictor implemented as a deep residual neural network with 220 residual blocks processing a representation of dimensionality $64 \times 64 \times 128$—corresponding to input features calculated from two 64 amino acid fragments" [49].

The original version of AlphaFold has 21 million parameters and the input for the network includes evolutionary profiles and co-evolution features, indicating that the training of this model requires substantial computer resources. Since distance map prediction is fundamental to protein structure prediction, a dataset to facilitate the testing and training of these maps has been developed [50]. Researchers may also benefit from AlphaFold2 by reducing the quality of the data that they need to experimentally gather to establish the structure of proteins. Eventually, as the protein structure–activity relationship is deciphered, it will be possible to test the structural model by performing an experiment to evaluate the predicted activity from the structural model.

It will be relevant to test the reliability of the point mutation models generated by AlphaFold2; this will determine the usefulness of this tool for protein activity annotation. For instance, many protein sequences that are very similar in sequence to others display different protein activities [51,52]; such is the case of paralogues and ohnologues that constitute difficult cases for protein activity annotation [53]. Another important problem of protein structural models is to differentiate the structures of proteins in isolation versus those found in complexes with other molecules. As noted in Figure 2, sometimes, proteins adopt large conformational changes in the presence of an interacting molecule or that are required for protein folding and activity. These changes are not always possible to infer from molecular dynamics simulations [54]. Since AlphaFold2 predicts one contact map that is relevant to the final protein structure, this last aspect of protein structure is unlikely to be properly modeled by that approach (see below). However, as more information about these large conformational changes becomes available, AlphaFold2 or similar approaches may be used to predict these conformations as well.

## 8. Protein Design

In the previous sections, I focused on the problem of predicting protein structure or activity given a protein sequence or structure, respectively. This section deals with the inverse problem, which represents an area of many health and biotechnological applications: to design the protein sequence that will fold into a particular structure or will perform

a particular activity. Effective protein design will allow us to obtain antibodies to target infectious diseases or to obtain enzymes with improved properties for biotechnological applications, among many others [55]. The design of protein sequences is fundamental to understanding the relationships described above between protein structure and activity. For instance, since a single protein sequence may display multiple protein activities and multiple sequences display similar activities, it is possible to envision a computer program that may design multiple protein sequences that would be compatible with a particular activity [56,57]. On the other hand, we noted that protein structure and activity may be modeled by a bijection; yet, proteins adopt multiple ensembles of protein structures and different ensembles are relevant for the different activities of a given protein. In this scenario, it seems reasonable to develop computational methods to predict multiple protein structures for a given protein. However, most methods to predict protein structure aim to predict the single, most stable conformation of the protein, which is similar to those reported in public repositories [58,59]; this is true even for the most advanced machine-learning-based methods [60,61].

Computer-based methods may overcome this limitation by effectively sampling the conformational space available for a given protein structure [62]. This ensemble of protein structures may be organized into subsets that contain similar protein structures as defined by a distance criterion. To achieve this, one possibility is to build residue contact maps using different distances; the larger the distance, the less sensitive the map will be to protein structural changes. Thus, the use of large distance cut-off values to build contact maps of amino acid residues may help to group many similar protein structures. For instance, contact maps derived from beta-carbon atom distances between 9 and 11 Å captured the most stable conformations of proteins, with up to 2 Å of root mean square deviation [63]. Thus, contact maps represent ensembles of protein conformations and can be used to study their relationship with protein activity [24] or to predict critical residues for protein activity [64]. This area deserves further exploration to promote the generation of a new class of computer methods that will predict the structural ensemble of a protein sequence associated with a protein's activity.

### 9. Conclusions

In summary, I have reviewed evidence showing that a single protein may display multiple activities as a consequence of having access to multiple protein structures either through allostery, post-translational modifications or other environmental factors. This indicates that protein sequence has a redundant relationship with protein activity, while protein structure is expected to present a one-to-one relationship with protein activity. Understanding the structure–activity relationships of proteins will improve our ability to selectively alter the mechanisms at play in biological systems as well as to design proteins. The solution to the problem of the protein structure–activity relationship resides in computational approaches and hence the future of biology, as has been the case in the past [65].

**Funding:** This research was funded by DGAPA-PAPIIT, grant number IT200320.

**Institutional Review Board Statement:** Not applicable.

**Informed Consent Statement:** Not applicable.

**Data Availability Statement:** No new data were created or analyzed in this study. Data sharing is not applicable to this article.

**Conflicts of Interest:** The authors declare no conflict of interest.

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
