# Peer review of "Challenges in the Computational Modeling of the Protein Structure—Activity Relationship"

_computation, doi:10.3390/computation9040039_

Round 1
Reviewer 1 Report
The Author's goal is to present the state of computational modeling of the protein structure-activity relationship, while focusing on the challenges still needed to be addressed.
This mini review is missing some points, that would be wellcome for better understanding of the topic.
- While there is no universal way to represent structure (S) and activity (A), it would help readers to at least include a reference to the most used representation, with mention as to what granularity (precission) is used for a representation. Without this, the bijection mentioned on line 148 is not valid and the relationship might not even be a function.
- When dealing with prediction methods, there are several deep-learning based methods that don't even get a mention. Also the inverse prediction, ie from a desired activity to a protein structure is not mentioned at all, although this has great practical value and is extensively researched.
I would support the publication of this paper if these points are addressed.
Author Response
This mini review is missing some points, that would be wellcome for better understanding of the topic.
1. While there is no universal way to represent structure (S) and activity (A), it would help readers to at least include a reference to the most used representation, with mention as to what granularity (precission) is used for a representation. Without this, the bijection mentioned on line 148 is not valid and the relationship might not even be a function.
I appreciate this comment. Indeed, there is no universal way to represent S and A, yet I did mention in the section entitled “Modeling protein structure-activity relationship” some of the common representations for both S and A. However, the granularity of such representations was not addressed and I agree that it is fundamental for the claim regarding the bijective nature of the structure-function relationship. To address this issue, I have added a new section about protein design and in there I included the following sentences to address this important aspect:
“On the other hand, we noted that protein structure and activity may be modeled by a bijection; yet, proteins adopt multiple ensembles of protein structures and different ensembles are relevant for the different activities of a given protein. In that scenario, it seems reasonable to develop computational methods to predict multiple protein structures for a given protein. However, most methods to predict protein structure aim to predict the single, most stable conformation of the protein, which is similar to those reported in public repositories [58,59]; this is true even for the most advanced machine-learning-based methods [60,61].
To solve this limitation, computer-based methods are able to sample the conformational space available for a given protein structure [62]. This ensemble of protein structures may be organized into sub-sets that contain similar protein structures as defined by a distance criterium. To achieve this, one possibility is to build residue contact maps using different distances; the bigger the distance the less sensible the map will be to protein structural changes, therefor many protein structures will group in the same contact map. For instance, contact maps derived from beta-carbon atom distances between 9 to 11 Å, capture the most stable conformations of proteins with up to 2 Å of root mean square deviation [63]. Thus, contact maps represent ensembles of protein conformations and can be used to study their relationship to protein activity [24] or to predict critical residues for protein activity [64].”
2. When dealing with prediction methods, there are several deep-learning based methods that don't even get a mention. Also the inverse prediction, ie from a desired activity to a protein structure is not mentioned at all, although this has great practical value and is extensively researched.
I agree that there are many methods to predict either S or A. It is not the intention of this review to refer to all methods available for those goals, yet I did refer to the world competitions that evaluate the best predictors for S and A, CASP and CAFA, respectively. Furthermore, I described one of the best methods to predict A according to CAFA (see reference 39) and I mentioned the best model to predict S according to CASP (see references 45 and 46). In this new version, I have added 10 new references that include deep-learning works related to protein structure and activity prediction.
Certainly, the problem of predicting S from A is a problem relevant for this review and I did not mention it. I appreciate this comment. To address this limitation, I have added a section entitled “Protein design”.
Reviewer 2 Report
the Del Rio manuscript is a good piece of work. But it should be improved, i.e. a clearer introduction, there is not a Conclusion, and also if it's possible to add more examples present in litterature.
Author Response
the Del Rio manuscript is a good piece of work. But it should be improved, i.e. a clearer introduction, there is not a Conclusion, and also if it's possible to add more examples present in litterature.
I appreciate the comments. To address these comments, that some of them were also pointed out by reviewer 1, I have added a section named Conclusion and added a section named “Protein design” to increase the number of examples present in the literature.
Round 2
Reviewer 2 Report
I think the manuscript should be accepted in this form.
Author Response
I appreciate for the comments that did help me to improve the quality of this work.